# Learning the Practice from the Practice: Theory–Practice Courses in Teacher Education

Orit Oved [1,*] and Nirit Raichel [2,3]

1   Faculty of Education and Teaching, Tel-Hai College, Upper Galilee, Qiryat Shemona 1220800, Israel
2   Department of Education and Community, Kinneret College on the Sea of Galilee, Tzemah 1513200, Israel; niritraichel@gmail.com
3   Graduate Department, Gordon College of Education, Haifa 3570503, Israel
*   Correspondence: ovedori@telhai.ac.il

**Abstract:** In teacher education programs, it is important to deepen knowledge alongside developing practices through practical experience. One practice of the Professional Development School's (PDS) model in clinical experience is designing courses linking theory to practice. The present study examines the perception of the administrative officers in colleges of education in Israel regarding the Ministry of Education's Theory–Practice (TP) courses and the challenges in implementing them in the curriculum. This study was conducted in 16 state academic colleges for education and 37 administrative personnel participated: presidents, vice presidents, rectors, deans, and heads of courses and practical training. A semi-structured interview was used, and the data was analyzed thematically. The research participants believe that TP courses as a tool implemented as part of the PDS model may be effective in training teachers to integrate theory with practical experience. The participants raised three major challenges to implementation: systemic, pedagogical, and organizational. The participants emphasize that designing and implementing TP courses is a complex, slow process requiring organizational change and the mindset of administration and teaching staff at both the colleges of education and the schools. Long-term assessment is required to examine the effect of reducing hours dedicated to education theory and subject knowledge.

**Keywords:** teacher education; theory–practice courses; clinical experience; practical teacher training

## 1. Introduction

Teacher education programs represent the first entry into the teaching profession. They include all the stakeholders: policymakers, teacher educators, schools and their educational staff, pre-service teachers, and the interactions between them [1,2]. Most models of teacher education include three components: (1) studies in the field of knowledge; (2) pedagogy—general methodology and disciplinary methodology (teaching the field of knowledge) [3]; and (3) practical training and specialization. In many education systems, this component is based on principles of clinical training [4,5].

One of the fundamental issues in the field of teacher training is the combination of theory (disciplinary and pedagogical knowledge) and practice in the teaching education program [6–8]. There is a gap between the reality in the schools and the teaching in the classrooms, on the one hand, and the theoretical principles taught in the teacher training programs, on the other [9–12]. An important part of teacher education is how teachers relate theory to practice and become experts in making decisions and implementing them while responding to pupils' needs [13]. It is very important to deepen the infrastructure of knowledge within the training processes and, at the same time, develop practices that connect knowledge and pedagogy and evaluate the connections between the two through feedback and reflection [14,15]. The practice component is also an important measure in establishing the feeling of readiness for teaching [16]. In-depth pedagogy studies and a rich

background of field experience significantly improve readiness for teaching and dealing with diverse groups [14,17].

The Ministry of Education (MoE) and the Council for Higher Education (CHE) in Israel published in 2020 a standards document that constitutes a reform in the field of teacher education. The document stated, among other things, that hours devoted to practical teaching experience should be expanded, according to the clinical experience model, by combining the material from the theoretical courses of the discipline/s and pedagogy with clinical training in the field. The standards document suggested that colleges of education design courses that combine theory and practice called theory–practice (TP) courses, based on the premise that practical teaching experience makes it possible to learn theory from cases in the field [18]. The purpose of this study is to examine the perception of the model of the TP courses and their implementation in the curriculum among holders of administrative positions in the state academic colleges of education in Israel. The research questions are as follows:

1. How do the educational administrators view the practice of the TP courses?
2. What are the challenges encountered in implementing the practice of TP courses?

### 1.1. Clinical Experience—The Professional Development School Model

Practical experience is an important and significant component in the training processes for the teaching profession and for working in complex settings, mainly because it is an integral part of the professional development of education students and because of its contribution to shaping them as professionals [19]. Training that focuses more on work in the classroom and provides pre-service teachers broad practical experiences and opportunities to explore and learn creates beginning teachers who are more effective and successful in promoting their pupils' achievements starting from their first year of teaching [20,21]. Therefore, in recent years, a key aspect in improving the quality of teaching and the quality of the teacher has been extending the period of practical experience and emphasizing the clinical aspect of experience, in an attempt to tighten the connection between experience and the theoretical courses to ensure that the teacher is guided professionally [22]. Better field experience occurs during the training itself and during the teachers' first year of teaching, under close supervision [4].

The clinical experience characterizes action-based professions, such as medicine or clinical psychology, and focuses on imparting professional skills at the same time as practical experience in the field. These experiences are an opportunity to learn about the profession from a reflective observation of the dialectic between theory and practice [4,23,24]. In this training, the need to move to practical experience in the field of teaching is emphasized, where learning processes take place in partnerships between the academic institution of education and the field [25,26]. Such clinical experience has a positive effect on the continuation of pre-service teachers as teachers of the future and invites a continuous evaluation of processes and practices among teachers as educators and among pre-service teachers [27,28].

One of the models that place the clinical experience at the core of the training is the partnership and fellowship models, such as the Professional Development School (PDS) model based on setting up schools for professional development [29,30]. The premise is that the university and the school are sources of knowledge and expertise, and the combination of the two is effective [31]. The model is focused on enriching theoretical knowledge, developing new teaching methods, applying them in practice, and exposing the pre-service teacher to extended practical experiences while being closely accompanied by the pre-service teacher mentor [24,32,33]. The PDS model features an emphasis on joint research processes and creating cooperative learning communities for teachers in the school, pre-service teachers, and instructors from the training institution. The goals are to improve teacher training by reducing the gap between theory and practice; provide professional development for all partners that is expressed in mutual empowerment on a professional, social, and personal basis; promote pupils in the school towards improving

their academic, social, and emotional situation; and finally, promote research processes to improve practice [34].

Studies show that pre-service teachers who learned using the PDS approach were found to be more prepared to work in the complex educational field [33,35,36]. However, there are challenges and barriers to the application of the clinical experience as well as difficulties that arise in teacher training, according to the cooperative model. Recruiting schools to the cooperative idea is based on volunteering; therefore, training institutions are forced to work with schools that are not necessarily suitable for providing such guidance, and that includes teachers who lack appropriate guidance skills [12]. Many reforms in the education systems do not leave the time and ability to promote partnerships with training institutions. The cooperative model involves the educational field's recognition of the benefits of cooperation with academia, a willingness to invest effort, relationships of trust, and the involvement of all participants. Furthermore, the partnership requires time and financial resources to reward teacher mentors and schools that are partners in teacher training with special and appropriate compensation [31,37].

### 1.2. The Context of the Research

Teacher education in Israel for state, state–religious, and state–Arab education takes place in academic colleges of education and schools of education in universities. The colleges of education in Israel are characterized according to the education system in the country and the various branches of education—state colleges, state–religious colleges, and state–Arab colleges. The state colleges of education train teaching staff, educators, and teachers for kindergarten, special education, elementary school, and schools up to the 10th grade, each according to the curriculum in use there. The general state colleges are intended for anyone interested in learning teaching without cultural or religious distinction, and they include pre-service teachers from the various sectors. The state–religious colleges mainly comprise pre-service teachers who belong to the religious sector and are interested in teaching in state–religious schools. In addition to the state standards, the curriculum also includes religious studies and methods for teaching them. The Arab colleges appeal to pre-service teachers from the Arab sector and some classes are held in Arabic in accordance with the curriculum of schools in Arab society in Israel. In all the state colleges of education, teaching certification is conducted in several tracks: teaching studies for a bachelor's degree (B.Ed) for school teachers and kindergarten teachers; in most colleges, a master's degree in Education (M.Ed) for those with a teaching certificate (teaching license); and programs for retraining academics for teaching and studies for a master's degree in teaching (M.Teach). The academic curricula of the colleges are built according to professional standards approved by the CHE. The teaching certificate is the same in all state colleges that are budgeted and supervised by the MoE and the CHE.

The most recent standards policy document, published by the CHE in 2020, adopted the recommendations of the standards report drawn up by a committee of experts to "examine the structure and outline of teacher training in higher education institutions in Israel" as a reform in the training of teachers and kindergarten teachers [38]. One of the principles of the report (Section 5) referred to the experiential component of the training— the clinical training and ways of connecting the theory to the educational field. In addition, the number of hours of practical clinical experience in the curriculum was expanded [18]. The assumption is that, as in other professions, clinical training has a central and essential role, and is the link between theory and practice and between studies and the educational field [39].

Furthermore, the standards document proposed that the colleges of education further strengthen the connection between theory and practice within the practical experience of the pre-service teachers in the schools by introducing theory–practice (TP) courses. The TP course comprises a theoretical component that includes several learning sessions with a lecturer (synchronous or asynchronous) and an experiential component to be used for practice and performing practical assignments related to the course material. The TP

courses expand the hours of practical teaching experience beyond what is required by the MoE teaching standard. The teaching approach is constructivist, meaning that the pre-service teachers are proactive in learning, and it allows them to evaluate the composition of the concepts and theories and their application to practical experience [40].

The colleges were offered two options for applying the TP course. One option is the formal model—student teaching two days a week at the same educational institution, accompanied by a pre-service teacher mentor and a teacher educator (i.e., pre-service teacher supervisor). On the third day, pre-service teachers perform tasks and assignments in the classroom according to the course syllabus. The second option is concentrated courses—one day of student teaching and the second day, attending two TP courses. The lecturers of the TP courses come to the school for the entire day of student teaching. Each time they visit a different school and accompany a different group of pre-service teachers, meeting with the whole group at the beginning of the day via Zoom and closing the day of student teaching with a Zoom meeting [40].

## 2. Materials and Methods

The research is based on a qualitative approach [41], which enables an in-depth understanding of the perception among various administrators in the academic colleges of education of the idea of the TP course as part of the PDS clinical experience sample in the training process and the challenges in its implementation. The subject of the current article is part of a broader study that examined the policy of the MoE and CHE concerning clinical experience in the academic colleges of education and the implementation of the reform that began in 2021.

### 2.1. Participants and Data Collection

The research was conducted in 16 state academic colleges of education and teaching (out of a total of 20 state colleges of education in Israel): seven colleges in state–general education, six in state–religious education, and three in state–Arab education. The interviewees were 37 research participants holding various positions in the colleges: 11 college presidents, four vice presidents, four rectors, five academic deans, and 13 heads of courses and practical training. The participants are the main stakeholders in the colleges that were studied to answer the research questions.

The presidents of the colleges of education are the ones who receive the standards document from the MoE and the CHE and are responsible for its implementation. The presidents are charged with outlining the vision, the organizational, and paramount peda-gogical goals according to the nature of the college and its needs. Eight of the 11 presidents interviewed are themselves senior professors in the field of education. Two presidents among the interviewees whose field of research is not education included in the interview the college rector or the vice president who specialize in education and are knowledgeable in educational issues. The deans and the heads of the tracks contributed in knowing the details of the practical implementation of the standards document, the challenges and barriers and how they are reflected in the curriculum with the teaching staff at the college, the training schools, and the pre-service teachers.

A semi-structured interview tool was used in the study [42]. A semi-structured interview was found to be most suitable for the research questions that we sought to examine [43]. The interview protocol was based on research literature on clinical experience and the PDS model. The interview questions relating to the TP course focused on three topics: their view of the principle of TP courses in the training process; the administrative and logistical organization of the various administrators in designing the TP courses at the college; and the challenges encountered in the implementation of the directive to integrate these courses into the curriculum.

*2.2. Research Process and Ethical Considerations*

Locating the participants was performed by a formal appeal by email to the presidents of the colleges of education. The message presented the purpose of the study and a request for participation while emphasizing the researchers' commitment to the anonymity and confidentiality of the interviewees, their position, and the name of the college. After receiving the college president's consent to take part in the study, a date was scheduled for an approximately 60 min interview using the Zoom program. The interviews were conducted between August 2022 and February 2023. Participation was voluntary and it was emphasized to the participants that they could withdraw at any stage during the interview and research. The participants confirmed their participation by email and at the beginning of the interview gave their permission to be recorded on a tape recorder. The interview protocol was sent in advance for the interviewees to read ahead of the interview. In some colleges, the president of the college asked to bring to the Zoom meeting additional administrators who are relevant to the subject being studied. In other cases, at the end of the interview, we asked the president for a recommendation for additional personnel whom they approved of contacting. This is how we made contact by email with functionaries at the administrative level: heads of courses and those responsible for the clinical experience at the college. The interviews were recorded on a tape recorder and transcribed. Since the presidents of the colleges and the administrators are known and recognized by name and position, and in order to maintain their confidentiality in the process of analyzing the data and presenting the findings, a number was registered to each person quoted and their position in the college and given to the interviewee (I) (for example, I-10.1, president). This study was approved by the second author's Institutional Review Board (IRB).

*2.3. Data Analysis*

The data collected from the interviews were analyzed using inductive thematic analysis [44]. The pattern, categories, and themes were constructed 'from the bottom up' by organizing the data into abstract divisions in three stages [45]. First, extracting categories of meaning from the data that allowed the identification of key elements to construct the categories; next, creating super-categories and sub-categories and the relationships between them [46]; and then, data-centered analysis, the construction of the theoretical explanation, and suggesting conclusions and implications for practice.

**3. Results**

In the analysis of the interviews conducted with the participants, two main categories were found: the first is the perception of the TP course model among the presidents of the colleges of education and those in administrative positions. This category refers to the TP model in teacher education from a conceptual aspect. The second category refers to the challenges in implementing the TP course model. This category refers to the difficulties encountered in the design stages of the courses and their implementation in the curriculum and in schools and the ways that administrators in the colleges deal with them.

*3.1. The Perception of the TP Course Model by the Presidents of the Colleges of Education and the Administrators*

This category refers to the essence of the model as a practice for improving the quality of teacher training by connecting the contents of theory and practice in an integrated course and the conceptual and professional dilemmas it raises.

3.1.1. A Positive Practice for Deepening the Practical Teaching Experience in the Training Process

All the participants have a positive view of deepening the practical teaching experience and promoting the idea of the TP course in college as part of the new reform. The participants who hold positions of vice president, rector, dean, and heads of programs recognize the advantages of the model and support the concept of expanding the clinical

experience in training that is in line with international trends. According to them, the TP course idea promotes the quality of teacher training so that the learning is more holistic and integrative, allowing the pre-service teacher a better overall understanding of the content of teacher education within the hours allotted in the curriculum.

> This connection between theory and practice through the fact that we introduce courses into the practical teaching experience and also integrate it into this experience is what creates a good connection for the pre-service teachers, and the practical training gains more depth. It becomes a bit sharper. But the number of hours has not changed (I-6.3., Head of elementary school track).

The participants noted that the model requires preparation by the college and new thinking about teacher training and an opportunity to refresh the teaching tracks, curriculum, lecturers, and courses, to update them and make them relevant. Designing a new TP course requires new thinking about the contents of the course, creating connections with other theoretical courses in the program and linking them to practice, bringing in lecturers who can teach in this framework, as well as defining goals and learning outcomes.

> This is the model that can bring the connection between the colleges, between academia and the field. [...]. This can also lead to second—third, and fourth—thoughts about the contents of the courses that the lecturers teach in the college, how suitable they really are for the practical teaching experience of the pre-service teachers in the schools. I think that throughout the years there was a disconnect between these two worlds, and the clinical experience from now on requires this connection between the two sides (I-14.4., coordinator of practical training).

Another advantage mentioned by the participants in the TP courses is the possibility of expanding the hours of clinical experience within the hours approved by the MoE for approval of studies and obtaining a teaching license: "The integration of theoretical courses with practical teaching is one of the ways to stay within the hours" (I-6.2., Rector).

Participants in the state–Arab colleges even described the respect and pride of lecturers from academia who take part in the school's educational activities. According to them, the school principals respect the college lecturers and treat them as experts in their field. The school principal and the teachers can use their knowledge, and they sometimes serve as consultants in solving educational issues.

> They are proud that they have lecturers and teacher educators and that they have pre-service teachers. The parents of the children are happy that they have more teachers in the school. They really see an increase in personnel and a team of teacher experts." (I-14.3., Supervisor of practical training).

### 3.1.2. Pedagogical Dilemmas

A fundamental dilemma, which arose chiefly among the senior administrators—presidents, vice president, and rectors—stemming from their vision of the College of Education and its unique character, relates to the identity of the College of Education as an academic research institution and its position among other academic institutions in Israel—universities and regional colleges. At the universities and regional colleges, the teaching is theoretical and does not incorporate practice, while the adoption and expansion of TP courses in the training process reduces theoretical teaching and increases the time for practical experience. There is concern among the College of Education officials that deepening practical experience at the expense of theory harms the college as an academic institution.

> Who are we? [...] A fourth-rate university? Or are we a quality teacher training institution? This identity affects what happens in our college in every area. We want to teach the academic subject properly, but the academic material is not the only component of the training process. There is a subject, there is training for teaching the subject, and there is the practical teaching (I-6.2., Rector).

Another issue that was raised by the interviewees related to the makeup of the study material (which lessons would be taught) in the teacher training program and the relative distribution between them: the connections between teaching theoretical knowledge and practical knowledge, the division into theory and practice courses and the connections between them, and the size of each component in the training process for effective training. College presidents and vice presidents referred to the issue conceptually, as an issue in the field of teacher training: "The delicate balance between the discipline and the teaching of the discipline is a central issue that we should emphasize in training processes" (I-6.1., president); while deans, heads of study tracks, and those responsible for practical training referred to this in practice, as part of their professional efforts in formulating a curriculum in the various tracks.

> A math lecturer can be excellent, and he will teach differential equations brilliantly and the pre-service teachers will understand what differential equations are, but this is not good enough for us. Because they don't understand what to do with it in the classroom and how to actually translate it to work in the classroom. [...] The division into classes on subject material and classes on methodology is a division that should be cancelled at some point (I-12.2., Supervisor of practical training).

However, there is an understanding by the participants that reducing theoretical content in a course also entails a theoretical loss, namely, the way of exposing the pre-service teacher to ideas and models that may be of help in the future: "It is something very beautiful, the TP course, but there is also a big concession here. When a pre-service teacher sits through 14 lectures in a classroom, he will have more knowledge than when he sits through four lectures and is ten more times in the field" (I-10.1, President). Moreover, the heads of the colleges do not have a well-founded assessment of the effectiveness of the change and whether it actually increases the quality of the training. The responsibility for examining the effectiveness of the model and its improvement lies with the college: "We have taken a very radical step here, but we need to see that it produces better teachers and not lesser ones" (I-10.1, President).

### 3.2. Challenges in Implementing the TP Courses

After the MoE's directive to integrate TP courses in the teacher training process, the participants presented challenges in implementing this within the college training program in terms of three aspects: systemic, pedagogical, and organizational. The challenges were expressed mainly among participants holding positions that are in practice involved in the design and running of TP courses, such as heads of tracks and those responsible for practical training. But also senior administrators, and especially vice presidents and rectors, presented challenges at the conceptual and systemic level in the college.

### 3.2.1. A Systemic Challenge

This challenge refers to the design of the TP courses and implementing them in a holistic view of all the stakeholders: the MoE and CHE, the college, and the schools who are training the student-teachers in the field.

At the level of educational policy-making—Organizing TP courses requires a change and the design of an integrated policy for all stakeholders who take part in planning and implementation: at the level of policymakers, at the institutional level (the college and the teaching staff), and at the school level (district, supervision, school administration, and educational staff):

> It's impossible to conduct [a TP course] if you don't build it systemically, if there are no connections between things. It's not just a point here and a point there. It's something that needs to have a systemic agreement, and it needs to be built in the way that collaborations take place (I-3.3., Head of teaching track).

Each of the stakeholders has needs, interests, priorities, and pressures, which do not coincide with the needs of teacher training institutions and conflict and make it difficult to organize TP courses and implement them in the field.

At the college institutional level—the officials in the colleges, in the positions of vice presidents, rectors, deans, and heads of courses, indicated that in the first step, the teaching staff at the college must be enlisted in the idea of the TP courses and emphasize its importance in the teacher training process: "You want committed heads of departments. You also need engaged lecturers. It's not easy" (I-15.1., Dean). All the participants stated that most of the lecturers do not readily agree to cooperate with the idea of the TP courses, both because they are not willing to teach in schools and because they are afraid of the reduction in teaching hours and theoretical content in the course.

> There are lecturers who refused to go there. They don't want to go to a school. They say, "I teach at a college, what do I have to do in a school?" That was a problem. So, we chose other lecturers who were willing to go there (I-14.3, Supervisor of practical training).

Since the lecturer cannot be forced to teach at the school instead of at the college, there is an understanding that the change among the lecturers must be a change in consciousness and take place over time.

> Do all the subject teachers go out to the field to see the pre-service teachers teaching? No. We don't obligate them, either. We really want it to happen, but we don't require it. [...]. I say this with regret, because I think it is very important (I-7.1., vice president).

According to the participants, implementing the idea of the TP course is a slow, delicate, and complex process. The change should be planned and modular and create collaborations in order to be significant and effective in the long term.

> Every year there will be another stage and that's fine. We are trying to expand the circles. So, we work through the department heads. First of all, we make clear to them the importance of this thing. [...]. This is really in-depth work on the curricula. (I-12.2., Supervisor of practical training).

The implementation of a TP course requires building a course by creating connections between different courses, preparing a syllabus and the course schedule, and dividing the hours between the lecturer, the didactic teacher, and the teacher educator. All the participants talked about setting up professional teams that include senior administrative position holders and heads of study tracks who have direct contact with the teaching staff at the college and a full understanding of the content of the track and its theoretical, pedagogical, and practical needs, and also limited and dedicated teams according to the study tracks. Some colleges even emphasized that they have defined a new role for organizing the courses and making the connections between theoretical learning and the field: "We work hard, we meet once every two weeks as a working team, and we think about connections, etc." (I-12.2., Supervisor of practical training).

### 3.2.2. A Pedagogical-Didactic Challenge

This category refers to the set of pedagogical and didactic considerations in putting together a TP course. As mentioned above, the participants noted that the decision to participate in a TP course rests with the lecturers, and courses are selected as a TP course only when there is cooperation by the lecturers. In addition, the courses selected are those that the school allows. So, the consideration for turning a regular course into a TP course is not necessarily the pedagogical and didactic consideration that this is necessary to improve the curriculum and training, but in some cases a question of what is there: "The teachers there are usually not purely subject teachers. Because a professor who teaches geography will not know how to conduct a TP course, so that is also taken into account" (I-13.3., Supervisor of practical training). Even when the lecturer responds positively and



starts teaching the TP course, "it doesn't really go smoothly. You need a whole system of monitoring the lecturers. They come, or they don't come to school, they conduct it properly, or they do not conduct it properly. It's very complex" (I-15.1., Dean). In some cases, the heads of the colleges compromise with the lecturers and find creative solutions, for example, "We use Zoom for the purpose; the lecturer makes visits to the schools and conducts plenary sessions by Zoom. We create some kind of mix here" (I-9.2., Rector).

Another problem is related to the scope of the theoretical material studied. Taking a theoretical course and turning it into a TP course using the PDS model causes a reduction in the scope of study subjects. The participants presented the claims made by some of the lecturers who taught the theoretical courses: "What is this? I will have four lectures instead of the fourteen I had until now. What did the education students leave with? What did they know? They know almost nothing" (I-10.1., president). Moreover, the topics in the course focus mainly on situations that arise in the school and on local interactions at a particular time. In this way, the pre-service teacher is not exposed to unfamiliar topics, which he is not even aware of but may need in the future during his teaching work.

### 3.2.3. Organizational Challenge

Most colleges of education in Israel are subordinate to two authorities, MoE and the CHE—two institutions whose requirements sometimes contradict each other and cause difficulties. While the MoE seeks to deepen the practice in teacher training, the CHE emphasizes the academic, theoretical aspect.

> There is a conflict of instructions here. According to the rules of the CHE, it is possible to teach up to a quarter of the degree outside of the college campus or in post-primary institutions that are not academic. According to the new outline, we can teach 4-6 courses in the schools, which is good news for us. Whether or not this work outs to about a quarter of the degree is another issue (I-9.2., Rector).

The CHE's requirement for academic institutions, including the colleges of education, to hold courses with many participants is also inconsistent with the characteristics of the TP course—up to 18 pre-service teachers per course: "You need courses of 50 or more. How can you take a course of 50 and make it a TP course?" (I-3.3., Head of teaching track).

Another organizational difficulty is related to teaching in the early childhood tracks and special education. According to the participants mainly in the positions of heads of tracks and those in charge of the practical training who are responsible for the relationship with the training schools and the placement of the pre-service teachers in the classrooms. In teaching tracks in kindergarten, special education, and education for pupils with multiple disabilities, it is not possible to hold TP courses because of the nature of the teaching in these classes and the structure of the educational institutions. In a kindergarten, there is but one kindergarten teacher, who cannot act as a kindergarten coach, instructor, and mentor for a large group of pre-service teachers. The physical structure of the kindergarten also does not allow for assigning a classroom for the pre-service teachers, so the studies must take place in the college.

Special education classrooms also do not allow the integration of TP courses. These classes are characterized by a relatively small number of pupils in the class. Moreover, not every school has a special education class; sometimes there is only one class in the school, so it is not possible to put many pre-service teachers in a classroom. This problem was also noted by officials from colleges that offer specialized courses in the field of art, subjects that are not in the core school curriculum or are optional subjects like drama, dance, sculpture, and painting.

The multiplicity of different functionaries visiting the school during the days of clinical experience also causes confusion. The introduction of subject lecturers into the schools in addition to the mentor teacher, the teacher educator, and sometimes also the didactic instructor, creates feelings of stress and confusion among the pre-service teachers. During the day of student teaching, they receive feedback from several functionaries, sometimes contradictory. The school principal and the educational staff also experience a deluge

of visitors to the school who interfere with the daily routine: "There are many visitors inside the school, and the principal and the pre-service teachers do not know who they belong to sometimes" (I-14.3., Supervisor of practical training). Additionally, there are not always suitable facilities in the school for teaching the courses, such as a spare room, a projector, Internet, and more: "Sometimes great ideas fail due to the lack of something technical, a teacher who isn't right for it, or a school that doesn't have computers and a room to assemble in, some small technical thing that is called "reality" (I-13.3., Supervisor of practical training).

## 4. Discussion

This study examined the perception of presidents of colleges of education and those with administrative and academic positions in 16 state colleges of education of the TP courses as part of the PDS model for deepening the clinical experience in the training process and expanding it within the curriculum. The research findings revealed two key dimensions in the participants' perception of the TP course in the clinical experience: The substantive dimension—relates to their perception of the TP course as a practice that enables a connection between theoretical courses (subject and pedagogical) and the practical experience, and the dilemmas it raises. The practical dimension—relates to challenges in the design of the TP course in the college curriculum and their implementation in practice.

The substantive dimension raised three main issues. First, all the participants recognized the integration of theory and practice as a central issue in the field of teacher training and the importance of the proportion between the curriculum of knowledge and practice, in light of the gap between learning in colleges of education and the complex reality of actual teaching in the field [7,47]. Examining education policy in 25 of the world's leading education systems, Barber & Mourshed [48] found that they developed teacher training programs that promoted a combination of theory and a building of practical skills. Linda Darling-Hammond and her colleagues' research [4], which examined teacher training policies in five countries with successful education systems, also revealed that successful education systems develop or expand partnerships between schools and universities, to provide practical experience that bridges theory and practice. Thus, the perception of the presidents and the various officials at the colleges of education is in line with international trends in the field of teacher training. The clinical experience edifies the pre-service teachers about the complexity of teaching and prepares them to cope better when they enter teaching.

All the participants had a positive view of the emphasis of the partnership model in the clinical experience and the advantages of the PDS model, which is focused on the enrichment of theoretical knowledge, its practical application in teaching practice, and the exposure of the pre-service teacher to extended teaching experiences while being closely accompanied by a mentor [24,49]. This model enables the implementation of the TP course and the expansion of teaching practice in the curriculum, as well as a holistic, integrative, and synergistic training process. The research participants believe that the TP courses foster a connection between the theory courses (disciplinary and pedagogical) and the practical experience in the curriculum for teacher certification. In this way, it is possible to present the relevance of the theory in the teaching work and reduce the gap between teaching in the college and teaching in the actual schools. The TP courses, like other case-based training methods that link theory and practice, such as case-based pedagogy [11], case study instruction [50], and case-based approach in teacher education [9], allow pre-service teachers opportunities to identify problems in their different situations, interpret them, and learn from events that occur in the school. This will enable pre-service teachers to analyze and think critically so they can make decisions to solve potential problems they face in the classroom. With the TP course method, the pre-service teacher can consolidate practical knowledge in an intelligent way that includes the theoretical knowledge and skills they have acquired. The practical knowledge will help them make effective decisions even in the first years of teaching when they are still lacking in professional experience.

A second issue deals with the relationship between practical knowledge and theoretical knowledge in teaching training. On the one hand, there is a recognition of the role of theory in the curriculum, as it plays an important role in teacher training and is an integral part of practice [15]. On the other hand, practical knowledge enables teachers to deal effectively with practical problems [3,51]. There is agreement among researchers that a balance needs to be found between the two types of knowledge [3,52,53]. Theoretical knowledge must be part of teacher training. It contributes to pre-service teachers seeing the big picture of educational knowledge. Furthermore, putting the emphasis on practical knowledge may potentially restrict the pre-service teachers' learning to the practical knowledge of their mentors if additional measures are not taken. However, sometimes the pre-service teachers need specific knowledge about a certain situation or problem they encounter.

A third issue, creating the connection between theoretical contents and their integration into practice during the pre-service teachers' clinical experience in the schools may reduce the theoretical contents that are taught in the college classes and focus mainly on the issues that arise during student teaching. Research by Rasmussen & Rash-Christensen [54] showed that problem-focused approaches are successful in the training process, and they narrow the gap between the theory taught in college and the actual teaching. Nonetheless, expanding the practice component raises several institutional dilemmas and fundamental pedagogical issues that policymakers and college presidents must resolve:

(1) The image of the College of Education as an academic institution and its standing among other academic institutions. The subordination to two authorities is also disadvantageous. In Israel, the degree awarded by the colleges of education, subordinate to the MoE's Department for Teacher Training, is the B.Ed or M.Ed, while the universities and regional colleges award B.A and M.A degrees, so that in any case their status is lower when compared to a university degree [55]. Integrating the TP courses, expanding practice, and reducing theory classes may harm these colleges' academic standing.

(2) The status of the lecturers in the colleges of education is an issue, both within the college between purely subject teachers who teach full theoretical courses in the college and lecturers of the TP courses who teach few theoretical lessons and whose teaching takes place mainly in the schools, and also in front of the professional teacher educators. Zeichner [34] believes that it is not easy to achieve change in the academic staff. In the PDS model, senior lecturers, school faculty, and education students should work collaboratively as equals. Moreover, not all lecturers are suitable for teaching that integrates practice. Questions arise about how should the college act in cases where a lecturer is not suited or not interested in teaching outside the campus. Should lecturers be required to?

(3) The issue of accountability in teacher education [49]. The College of Education is responsible for the training of teaching staff in the schools in the long term. The training should be comprehensive and professional and lead the pre-service teachers to succeed in their jobs. The curriculum should present a broad picture of the training content, including exposure to ideas and situations that are unfamiliar or unknown to the student, ideas, and situations that they have not yet encountered but for which must be prepared because they may encounter them in the future. Zeichner and Bier [26] also point out that in training based on local school scenarios, there is concern about a technical focus on teaching skills and less on the teacher's education and education to multiculturalism.

(4) The effectiveness of the TP course has not yet been measured both in terms of the effectiveness of expanding practice over theory and how to integrate them and in terms of the pre-service teacher's sense of readiness in the first years of their teaching work. Studies show the advantage of clinical training in the PDS model. But integrating the TP courses into the curriculum and their effectiveness in the field of training in the immediate term (readiness and competence during the training process and upon its completion) and in the long term (in the teacher's first years of teaching) have not yet been investigated. It is important to conduct an assessment of the TP courses among all the parties that have a share in their implementation and operation: the college, the pre-service teachers, and the



training schools, in order to prove their effectiveness and to continue to have them in the curriculum in the colleges of education.

There is no agreed answer to these dilemmas and the questions they raise at this stage. A focused evaluation study of the TP course is required regarding the college-organizational aspect and the teacher's professional ability.

The practical dimension presented three major challenges in the design and implementation of the TP courses in colleges of education: systemic, pedagogical, and organizational. Senior administrators holding the positions of college presidents, vice presidents, and rectors presented challenges mainly from the perspective of the college's vision and its unique character, while those in positions such as deans and heads of tracks and practical training elaborated in their description of the challenges and barriers they encounter as part of their work in designing and implementing TP courses. In an examination of the challenges revealed by this study, four basic principles emerged for the effective integration of theory and practice in the teacher training program and for the success of the model and its long-term implementation.

(1) Teacher education reform requires cooperation between all stakeholders: the educational policymakers, the colleges of education (those in administrative positions, teacher educators, teacher educators, and pre-service teachers), and the schools (the school administration and teacher mentors). Teacher education reform can only be effective if the policy is well implemented. Support for reform requires a coherent framework and the ability to evaluate and implement at all levels of the education system [56]. The directive of the policymakers to create TP courses within the teacher training should relate to policy design for educational institutions as well. The guidelines should not be contradictory but supplement each other.

(2) Cultivating a positive perception of the contribution of TP courses in the training process—both for the teaching staff in the colleges to increase the motivation for cooperation and to the administrative and educational staff in the schools so that they cooperate both in human resources, such as including quality training teachers who can be learned from. Zeichner [34] notes that in his experience, the change is slow but necessary for the PDS model to succeed. According to him, in the US, teachers tend to reject or ignore the expertise of university professors, and often professors believe that their only role is to disseminate knowledge to teachers. Acknowledging the expertise of each of the parties may increase the motivation for academic and professional collaboration between colleges and schools.

(3) Establishing teams at different administrative levels: presidents and rectors who outline the institutional policy; heads of study tracks who are the most familiar with the study program in the track, the contents of the courses (the syllabus), and the lecturers who teach there. Working out the TP courses in teams with the maximum participation of the stakeholders in the college will lead to the development of a plan that is built on feedback and trial and error. In this way, from one year to the next, the TP courses can expand to more courses in the various tracks.

(4) The considerations in choosing the courses that will be used as TP courses integrating theory and practice should be based on pedagogical and didactic considerations in the teacher training process. Which courses can be taught in the schools and incorporate the practice? What is the ratio between the theory classes and the practical experience during the course? Who will be the main guide in providing feedback to the pre-service teachers: the lecturer? the teacher educator? the mentor? These questions and more are important in the design stages of the course at the colleges of education.

## 5. Conclusions and Implications for Practice

The research participants in the various positions, from the president of the college to those in charge of practical teaching experience, believe that the TP courses as a tool that can be implemented as part of the PDS model may be effective in teacher training that combines the theory in the curriculum with the actual teaching experience in the schools. According to this model, the pre-service teachers gain first-hand experience in

the schools for two to three days, when connections are made between the theoretical content and the situations they encounter during the teaching. The pre-service teachers are accompanied by a supportive framework provided by the course lecturer, the teacher educator, and the mentor, providing feedback to the pre-service teacher from different perspectives. The analysis of situations in the field in real-time underscores the relevance of the theory to the teacher's work in the present as a pre-service teacher and in the future as a professional teacher.

However, the challenges mentioned by the research participants emphasize that the design of a TP course and its implementation is a very complex, slow process that requires a change in the mindset of the stakeholders: the administration and teaching staff at the colleges of education and the schools themselves. The teaching staff at the colleges must be enlisted to the principle of connecting theory and practice in the training process. On the one hand, they must be persuaded of its contribution to the pre-service teacher's readiness and competence in their teaching work and their professional development and, on the other hand, also to the lecturers themselves, who demonstrate the relevance of the theory taught to the actual work in the schools. As for the schools, the administration and teaching staff must be recruited to understand the importance of the idea of the TP course for the field of teaching in general and for professional teachers in particular. They, for their part, will support the idea by providing infrastructure, such as study rooms, internet, etc., while the contribution of the MoE is to allocate a budget as proper compensation for the schools and the mentor to increase their motivation to cooperate.

Another change that must take place is the organization of both the college and the school. It is recommended that a dedicated team be established that includes administrative positions and teacher educators for cooperation in developing TP courses. The staff will take action, based on the theoretical and practical needs in the various teaching tracks and in an informed process through feedback from all stakeholders (including a teacher educator, mentor, pre-service teacher) to improve the model and expand it to additional courses. In addition, they will formulate training programs for lecturers in building TP courses and teaching courses within the school to get to know and adapt to a different way of teaching and to know optimally how to link theory with practical tasks.

And finally, an accompanying evaluation is required for the process of designing TP courses and implementing them in the training program to examine the long-term efficacy in reducing theory hours. The TP courses will be able to survive and serve as an effective tool for connecting theory and practice in the long term only if they prove their efficacy in training quality teachers for the education system. It is important to have an institutional assessment while implementing the TP courses among all the partners: the administrative staff at the colleges and the heads of the tracks, the teachers, pre-service teachers, the schools, and the teacher mentors. This evaluation will help identify the strengths, challenges, and barriers to implementing the TP courses in the training program. In this way it will be possible to introduce changes, improve and refine their operation, and increase their effectiveness in accordance with the goals and objectives—strengthening the connection between theory and practice. Action research can be encouraged both among the college teachers themselves and in pre-service teachers' research courses at the college, which will be a tool for institutional learning and as a case study. Through such research, the chances may be increased that teacher training can make a difference in the long term and also raise the profile of the colleges of education as academic institutions and establish their position among other academic institutions.

In addition, it would be significant to conduct an evaluation among teachers in the first years of teaching who studied in TP courses and to evaluate in the contexts of self-efficacy, professional efficacy, and long-term stability in the teaching profession. The evaluation of the TP courses will be used by both the policymakers in the MoE and the colleges of education in delineating an effective curriculum and promoting teacher quality and teaching quality.

**Author Contributions:** The authors contributed to the manuscript equally. Conceptualization, O.O. and N.R.; Methodology, O.O. and N.R.; Data curation, O.O. and N.R.; Writing—original draft, O.O. and N.R.; Writing—review and editing, O.O. and N.R. All authors have read and agreed to the published version of the manuscript.

**Funding:** This work was supported by the Mofet Institute, a center for the research, curriculum and program development in teacher education, Israel.

**Institutional Review Board Statement:** The study was conducted according to the guidelines of the Declaration of Helsinki, and approved by the Institutional Review Board (or Ethics Committee) of Gordon College of Education, Israel (protocol code 2021.006, 30 November 2021).

**Informed Consent Statement:** Informed consent was obtained from all subjects involved in the study.

**Data Availability Statement:** Anonymized data are available upon reasonable request from the corresponding author.

**Conflicts of Interest:** The authors declare no conflicts of interest.

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
