# Peer review of "Learning the Practice from the Practice: Theory–Practice Courses in Teacher Education"

_education, doi:10.3390/educsci14020185_

Round 1
Reviewer 1 Report
Comments and Suggestions for Authors
I believe this is an important look at a complex issue. You have done a commendable job unravelling a multi-faceted endeavor. I would like to see a little more reference to the number of participants who shared your major points. There are a few references to "all,"but it seems like there might be differences between categories of roles. Please address this in your narrative.
It took me a while to understand that these colleges only prepare teachers and have no other programs. I would recommend adding a little more context for the colleges so it is clear why presidents would be important contributors. This would rarely be the case in a US college, as we no longer have single purpose colleges.
There are also places where it is not clear who you are referencing when you use "students." It eventually became clear that you only address college students, preservice teachers, but there are a number of places that one could read "students" as K-12 students. e.g. Line 84
I appreciate that you have included sources that are considered the experts in this area along with fairly current studies.
You might emphasize even more than you have how crucial it is to evaluate the efficacy of this model. It will not survive if not proven highly effective in preparing excellent beginning teachers.
Comments on the Quality of English LanguageGenerally the English is good. I would recommend that a critical editor give it a deep and thorough read. There are a few small things that occasionally detract from the flow of the narrative.
Reviewer 2 Report
Comments and Suggestions for Authors
The document is skillfully crafted and presents a clear and coherent narrative. It adheres to established standards, and I endorse the publication of the study. Nevertheless, I have a few minor suggestions for the authors to consider incorporating.
It would be beneficial to include more information about the overall status of state colleges, state religious education, and state Arab education. This would provide an insightful overview for the international audience, highlighting the distinctions among the three types of state colleges.
Introducing sample questions for semi-structured interviews and including DOIs for the references would enhance the completeness of the manuscript.
